# Autoformalization with Large Language Models

**Yuhuai Wu**[1,2†]  **Albert Q. Jiang**[3]  **Wenda Li**[3]

**Markus N. Rabe**[1]  **Charles Staats**[1]  **Mateja Jamnik**[3]  **Christian Szegedy**[1]

[1]Google Research
[2]Stanford University
[3]University of Cambridge

## Abstract

Autoformalization is the process of automatically translating from natural language mathematics to formal specifications and proofs. A successful autoformalization system could advance the fields of formal verification, program synthesis, and artificial intelligence. While the long-term goal of autoformalization seemed elusive for a long time, we show large language models provide new prospects towards this goal. We make the surprising observation that LLMs can correctly translate a significant portion (25.3%) of mathematical competition problems perfectly to formal specifications in Isabelle/HOL. We demonstrate the usefulness of this process by improving a previously introduced neural theorem prover via training on these autoformalized theorems. Our methodology results in a new state-of-the-art result on the MiniF2F theorem proving benchmark, improving the proof rate from 29.6% to 35.2%.

## 1 Introduction

*Autoformalization* refers to the task of automatically translating from natural language mathematics to a formal language [46, 42]. The implication of a successful autoformalization tool is huge in both practical and philosophical terms. It would reduce the currently excessive cost of formalization efforts [27], and in the long-term it could connect the various research fields that automate aspects of mathematical reasoning, such as automated theorem proving and computer algebra, to the vast body of mathematical knowledge exclusively written up in natural language. Moreover, autoformalization would be a true testament to machine understanding, grasping both the fuzziness of natural language and the preciseness of formal language.

Recent advances in large language models [8, 10] showed promising capabilities of understanding formal languages [9, 32]. However, the existing successes are limited to formal languages where there exists a large body of corpus on the web (e.g., Python language). Formal mathematics data is very scarce. For example, one of the largest formal mathematics libraries, the Archive of Formal Proofs, is only 180MB in size, that is less than 0.18% of the training data for the large language model Codex [9]. Moreover, unlike in the case of commonly used programming languages, where natural language docstrings are broadly available, there is almost zero aligned data between natural language and formal mathematics. Therefore, it is unclear the recent successes can directly contribute to the development of autoformalization.

In this work, we explore the prospects of autoformalization with large language models. To our surprise, we find that large language models already have a decent capability of formalizing natural

---

[†]Corresponding author (`yuhuai@google.com`).

36th Conference on Neural Information Processing Systems (NeurIPS 2022).

Figure 1: Case study 1: An example of a perfect translation from natural language to Isabelle code.

language mathematics in an interactive theorem prover. See Figure 1 for a perfect autoformalization example. The model not only translates into syntactically correct Isabelle code, but also grasps the non-trivial reasoning in natural language. We randomly pick 150 formalizations and manually evaluate their correctness. Among them, LLMs are capable of producing 38 perfect formalizations! As an application, we further demonstrate that autoformalization can provide useful training data for neural theorem provers. We use autoformalized statements as targets for proof search with a neural theorem prover for Isabelle/HOL. After fine-tuning our neural theorem prover on the proofs it found, its success rate on the MiniF2F benchmark [52] increases significantly, achieving a new state-of-the-art result of 35.2% theorems proven.

## 2    Related Work

Early applications of machine learning in theorem proving include the works by Schulz [40] and Urban [43], and later, directly guiding interactive proof assistants using machine learning techniques [15]. The revolution of deep learning then kicked off a new wave of interest in the topic starting with DeepMath [1, 33].

Several approaches have been suggested to address data scarcity: Imitation-free reinforcement learning was used to avoid the need for training on human proofs [31, 6, 15, 49]. Also, hindsight experience replay [2] was used to generate additional training data [5]. Hahn et al. [19], Schmitt et al. [39], Kreber & Hahn [29] and Wu et al. [50] have shown that training on synthetic formulas can be successful for temporal logics and inequalities. Rabe et al. [37] masked out different subexpressions from formal mathematical statements and generated 100 training examples for each source statement. Skip-tree data can also be used to improve the performance of neural theorem provers [22].

Wang et al. [46] explored the use of supervised and unsupervised translation techniques for autoformalization. Supervised translation yielded interesting results, but relied on synthetic (natural-looking) data that was generated by the Mizar theorem prover, while we rely on models trained via self-supervised language modeling, not trained for this particular purpose.

## 3    Background

**Formal Mathematics**    A few important and complex results of mathematics and computer science have been formalized manually using *interactive theorem provers*, such as the four color theorem [16], the Kepler conjecture [20], the odd-order theorem [17] and the verification of a microkernel [27]. This gives us almost complete certainty about the correctness of proofs, which can be of great value to resolve doubt about the correctness of complicated mathematical proofs or proving certain properties of software used in safety-critical applications, such as aircraft components [28].

These projects relied on interactive theorem provers, such as Isabelle [48], Coq [12], HOL Light [23], and Lean [13], which are essentially programming languages that enable users to enter their statements and proofs in a formal language, and which can then be checked automatically for correctness. Interactive theorem provers offer a limited amount of automation, but projects that formalize complex problems typically span many years of tedious work by specialists. Only in narrow domains like chip

design and the verification of drivers in operating systems has the automation of logic made sufficient progress to find commercial applications.

Progress in autoformalization and the automation of proofs might eventually make mathematics a universally available tool and enable a paradigm shift in science and the development of (safety-critical) software. Our interest in formalizing mathematics, however, has an additional aspect. We believe that autoformalization will serve a dual purpose and will not only accelerate the development of tools for mathematical reasoning, but also provide a means to ground machine learning systems, enabling a positive feedback loop between machine learning and formal systems (cf. [42]).

**Large Language Models**  Our work relies heavily on large language models (LLMs), in particular on PaLM [10] and Codex [9]. The training goal of these models is to predict the next word given some prefix. This allows us to train these models on arbitrary text, which is available in vast quantities. After training the models on hundreds of billions of words (cf. [25]), they are often able to generate high-quality text. We can also give these models an arbitrary prefix (the *prompt*) that they are then supposed to continue, which gives us some control over what they generate. This has been demonstrated with news articles, conversations, summaries, jokes, and poems. LLMs have also been evaluated on natural language word problems on datasets such as GSM8K [11] and MATH [24], and have been shown to make progress on these benchmarks with increasing scale [10].

**In-context Learning**  Large language models have shown a remarkable ability to learn patterns and tasks within the current input (context) that they are given [8]: this is called *in-context learning* or *few-shot learning*. For example, if we prompt a language model with a few pairs of English and matching French sentences, and end with a new English sentence, then the language model is very likely to pick up on the translation task and attempt a translation of the last English sentence. This observation has been used, for example, to achieve strong translation performance without access to large corpora of matching sentence pairs [21].

This allows us to specify the task of autoformalization simply by giving a couple of example formalizations. In Section 4 we will detail how exactly we use in-context learning for autoformalization.

## 4  Autoformalization for Mathematical Competition Problems

Inspired by the success of LLMs for synthesizing computer code by co-training on both natural language and code on web-scale data, we explore the capabilities of LLMs to turn natural language mathematics into formalized theorems for the interactive theorem prover Isabelle. This can be seen as a machine translation task (cf. [47]) in which the input language is English and output language is formal code used by the interactive proof assistant Isabelle [48].

We first study autoformalization in a constrained setting – formalizing mathematical competition problem statements. This setting has the advantage that most of the required background theory and definition has been formalized in the current libraries of Isabelle, so that formalizations are often possible without introducing additional definitions.

We start assessing LLMs' abilities to do autoformalization with a case study. We manually pick two interesting natural language mathematical statements, and prompt PaLM models of various scales [10] as well as Codex [9] to translate them into a formal statement in Isabelle. Next, we study a dataset in which we have human ground truth formalizations. The dataset is a subset of the miniF2F [24] dataset consisting of 140 algebra problems and 120 number theory problems. Using human formalizations as the reference, we compute the BLEU scores of the formalizations produced by several LLMs. Lastly, we perform human evaluations on failure cases in autoformalization on 150 problems.

Note that many mathematical competition statements are often of the form in which one asks to find the answer to a certain problem, instead of *proving* a given proposition. However, formal mathematical statements are in the form of propositions, instead of questions.

To transform a question into a proposition, we append the final answer after the question:

$$\texttt{\$Problem\_Statement}\ \textit{The final answer is}\ \texttt{\$Answer}.$$

The format of the prompt we use to do autoformalization is:

$$\text{Natural language version: } \texttt{\$Natural\_Language\_Statement}.$$
$$\text{Translate the natural language version to an Isabelle version:}$$

### 4.1 Mathematical Competition Datasets

**MATH** [24] contains in total 12,500 (7,500 training and 5,000 test) middle school and high school mathematical competition problems. Problems are taken from past mathematical competitions, including AMC 10, AMC 12, AIME, and more, and many can be found at `http://aops.com/community/c3158_usa_contests`. The dataset contains seven categories: `algebra`, `pre-algebra`, `intermediate algebra`, `number_theory`, `precalculus`, `probability`, `geometry`. Problem statements are written in LaTeX.

**MiniF2F** [52] is a recently introduced benchmark containing 488 mathematical competition statements manually formalized by humans in three different formal languages. Its goal is to compare and benchmark methods across different theorem provers for machine learning research. Some of these problems come from the valid and test set of MATH `algebra` and `number_theory`, and others come from previous International Mathematical Olympiad competitions or AoPS[1]. Note that the Isabelle formalizations of the miniF2F benchmark were committed to the repository during March, 2022. According to the public information of the training data, we think it is highly unlikely these formalizations were included in the pre-training corpus.

### 4.2 Case Studies

**Experimental setup** For all our experiments, we use the standard greedy decoding (i.e., temperature 0, $p = 1$) to obtain the autoformalizations. We randomly select two mathematical statements for constructing the prompt, which we provide in Appendix A.1. That is, no prompt engineering / tuning is performed when constructing the prompt. The natural language problem statements used in the case studies are taken from the miniF2F dataset. In the case studies below, we highlight the output of language models in red to distinguish it from the prompt.

**Case Study** 1 (**Figure 1**) We study the example shown in Figure 1, in which we ask LLMs to autoformalize an International Mathematical Olympiad problem[2] in natural language. Surprisingly, Codex is able to autoformalize the natural language statement as an Isabelle theorem perfectly, with output given. This is surprising for the following reasons.

First of all, the amount of Isabelle code is very scarce on the internet. The entire AFP library, the largest formal library that contains most of Isabelle proofs, is only 180MB in size. Even assuming that all of this data was included in the training of Codex, this makes at most $0.18\%$ of the pretraining data on which Codex was trained. The fact that the model can write syntactically correct Isabelle code at all is already fascinating.

Second, there is almost zero aligned data from natural language to Isabelle on the web. While some Isabelle files have comments, they typically only give a very high level description of what the theory being formalized is about. So either LLMs are able to transfer knowledge quite successfully between natural language and formal mathematics, or the task was learned mostly via few-shot learning.

Last but not least, the model is capable of understanding and formalizing nontrivial reasoning. First, the model is able to formalize the non-existence statement via proof-by-contradiction. To formalize "there is no function $f$...", it assumes there is such a function, and aims to prove "False". Second, the model understands what it means by the phrase "to itself", and correctly infers the domain of function: `f :: "nat \<Rightarrow> nat"`.

On the other hand, PaLM made some syntactic mistakes while getting most of the structure of the proof correctly, with outputs shown in Appendix C.1.

---

[1] `https://artofproblemsolving.com/`
[2] A problem from IMO 1987.

Figure 2: Autoformalizations from natural language to Isabelle code. **Left:** Case study 2 – perfect formalization by PaLM. **Right:** Case study 3 – incorrect formalization by Codex.

**Case Study** 2 **(Figure 2)**  In the next example, we ask LLMs to autoformalize a grade school mathematical word problem. Remarkably, PaLM and Codex are both capable of formalizing the statement perfectly. This is surprising because formalizations of grade school math problems in interactive theorem provers are rare (if they exist at all), as this type of mathematics is not of interest to formal mathematicians. Even more, none of the examples in the prompt (see Appendix A.1) that we provide are of this type. It is hence remarkable that the model is capable of extrapolating to this type of statement, showing a great promise of using LLMs for autoformalization.

To study this problem in more depth, we probe PaLM models of various sizes (8B, 64B, 540B) with outputs shown in Appendix C.2, and notice that scale is crucial for the LLMs ability to formalize. We observe that the 8B and 64B models are incapable of formalizing this problem, but the largest 540B model is able to produce a correct formalization.

**Case Study** 3 **(Figure 2)**  In our third case study, Codex gives an incorrect formalization in Isabelle. The mathematical statement involves a concept of "linear function", which the model fails to formalize correctly. Codex assumes this is already a known concept in Isabelle, and made up a name: `linear f`. Can the model learn to formalize such problems if the prompt contains an example that explains the concept of a line? We explore this and give an affirmative answer to the question (see Appendix C.3). Once seeing a tangentially related problem that explains the concept of a "line", Codex is able to perfectly formalize a "linear function". This shows the importance of the few shot examples we include, and also how good a few-shot learners these models are!

**Has the model memorized these formalizations?**  Whilst we do not have access to the training set of Codex, we attempted to find any occurrences of the formalizations produced in the case studies on the internet. We Googled them in different variants and inspected the first page of the search results. We tried variants with and without an "Isabelle" prefix, with and without quotation marks and other special characters, and also individual parts of it, such as "Isabelle "n mod 8 = 7"", but we did not find any occurrences of related statements. We also tested that we are indeed able to find occurrences of Isabelle formalizations on the web with this methodology, using pieces of formalizations picked from several websites, including the Archive of Formal Proofs. Hence, we are confident that the model has not memorized the formalizations it generated.

## 4.3   BLEU for Model Comparisons

The miniF2F benchmark contains 140 algebra problems and 120 number theory problems from the MATH dataset. For these problems, we have human ground truth formalizations in Isabelle, which gives us an evaluation set with pairs of natural language statements (from MATH) and their formalizations. We use this dataset to quantitatively compare different LLMs.

Table 1: BLEU scores between the autoformalized statements and human formalized ground truth.

| Models \ Subject | algebra | number_theory |
|---|---|---|
| PaLM 8B | 31.49 | 22.10 |
| PaLM 64B | 43.13 | 31.43 |
| PaLM 540B | 50.30 | 36.16 |
| Codex | **57.13** | **43.33** |

Table 2: Failure case study of 150 problems formalized by Codex.

| Failure cases \ Subjects | algebra | number_theory | inter_alg |
|---|---|---|---|
| Perfect translation | 13 | 17 | 8 |
| Incomplete/ill-formed/unclear prompt | 9 | 3 | 14 |
| Fail to align definitions or concepts | 10 | 18 | 18 |
| Inconsistent/missing assumption | 8 | 9 | 9 |
| Syntactical/type error | 7 | 2 | 11 |
| Missing definition in Isabelle | 0 | 12 | 3 |
| Wrong application of functions | 6 | 13 | 16 |
| Other | 6 | 2 | 1 |

Given the observation about few shot learning in Case study 3, we decided to add more relevant examples to each subject to improve the quality of autoformalization. For each subject (i.e., `algebra` and `number_theory`), we randomly sample 10 problems to construct the few shot prompt. The rest of the problems are used for evaluation (i.e., 130 for `algebra` and 110 for `number_theory`. We provide the prompt used in the Appendix B.1 and B.2.

We use PaLM models of varying sizes and Codex to perform the autoformalization, and compute the BLEU scores of the formalizations, shown in Table 1. Confirming our observation in Case study 2, we see a clear trend that scaling improves translation, as the BLEU scores consistently improve when we scale PaLM models from 8B to 540B, for both subjects. In addition, we see that the Codex model is better at autoformalization measured by BLEU, possibly due to the fact that Codex was trained on more formal data than PaLM.

### 4.4 Human Evaluation of Failure cases

To better understand LLMs' ability to do autoformalization, we manually inspect Codex's autoformalizations of 150 random problems from the MATH dataset [24]. 50 of the problem statements are sampled from the `algebra` training set, 50 from `number_theory` and 50 from `intermediate_algebra`. For `algebra` and `number_theory`, we use their corresponding prompt as in the last section, shown in Appendix B.1 and B.2. For `intermediate_algebra`, we use the prompt we used for `algebra` (Appendix B.1). We classify the failure modes of these translations, shown in Table 2.

We see that out of 150 problems, Codex is capable of translating 38 problems perfectly – a success rate of 25.3%. The majority of the failures are due to the misalignment of informal and formal definitions. For example, when seeing the phrase "the greatest possible value", the LLMs often fail to align it with the function `Greatest/Max` in Isabelle. Another example is the failure to align the factorial of $n$ (i.e., $!n$) to `fact n` in Isabelle. Other common failure modes include the misapplication of functions (e.g., applying a prefix function in an infix way).

## 5 Autoformalization for Neural Theorem Proving

To demonstrate the usefulness of the formalized statements, we explore if one can improve neural theorem provers by training the neural models on proofs of automatically translated theorems. In this section, we combine autoformalization with expert iteration algorithms [4], and achieve a new state of the art in miniF2F benchmark.

## 5.1 Expert Iteration with Autoformalization

The basic idea of expert iteration [4] is to iteratively generate a better dataset using the model, and use the data to improve the model quality. This allows the model to generate an even better quality of the dataset and hence a better model, forming a self-improvement cycle.

In neural theorem proving, one way to get better quality data is to use feedback from the proof checker to run many proof searches (or generate multiple proofs) and check the proof attempts for correctness. Newly found correct proofs can then be used as the new training data to improve the neural prover [7, 35, 36]. The main critical ingredient that is needed is a set of problem statements on which the model can perform proof search to obtain new training data. However, unlike in Polu et al. [36], where one asks humans to manually formalize a set of problems to get formal statements, here we use LLMs to autoformalize the theorems in order to kick off the self-improvement cycle.

More formally, denote a base neural theorem prover as $M_0$. Let the set of autoformalized problems be $\mathcal{A}$. For each iteration $i = 1 \ldots N$, we carry out the following procedure: use the language model $M_{i-1}$ with best-first search to prove as many theorems as possible in $\mathcal{A}$, collect the set of successful proofs $S_i$, concatenate successful problems from all iterations with the formal mathematics problems to create the set $\mathcal{A}_i = (\bigcup_{j \leq i} S_i) \cup \mathcal{B}$, and fine-tune $M_0$ on it for exactly one epoch to get a new model $M_i$. When we take the union of successful proofs from all past iterations, we perform deduplication by problem statements, similar to Polu et al. [36].

## 5.2 Neural Theorem Provers

To demonstrate the effectiveness of the approach, we start with a recently introduced neural theorem prover for Isabelle, LISA [26]. The LISA agent is fine-tuned on the PISA dataset [26] (extraction and interaction code under a BSD license), which consists of 2.49 million proof steps from the Isabelle/HOL library (under a BSD-style license) and the Archive of Formal Proofs (under various licenses as described here). The model is trained with the objective to predict the next token in a proof step, given the proof state and the last proof step. Following the setup of Thor [3], which achieves SOTA performance in the no-additional-training-data category on MiniF2F, we invoke Sledgehammer with a 30 second timeout when the model generates a step containing any of the keywords `metis`, `meson`, and `smt`.

We use Wang [45]'s implementation (under an Apache license 2.0) of a GPT-2 [38] style decoder-only transformer [44] model with 700M non-embedding parameters. The model has 24 layers, 24 attention heads, a hidden dimension of 1536, and a vocabulary size of 50400. We pre-train the model on the GitHub + arXiv subsets of The Pile [14] for 200,000 steps, with a context length of 2048 tokens. In pre-training we use a warmup strategy [18], raising the learning rate linearly from 0 to $2 \times 10^{-4}$ in 3,000 steps. We then use a cosine learning rate scheduler [34] for the rest of the pre-training, with a final learning rate of $1.2 \times 10^{-5}$. We use a global batch size of 32 sequences, or 65,536 tokens. For fine-tuning we use the same learning rate schedule, with 10,000 warmup steps, 90,000 annealing steps, maximum learning rate $3 \times 10^{-4}$ and final learning rate $3 \times 10^{-5}$. The global batch size is 144 sequences, or 294,912 tokens. The model's evaluation loss reaches a minimum after 13,000 steps and we use that checkpoint. For the fine-tuning in expert iteration, we fix the learning rate at $3 \times 10^{-4}$ and the batch size at 144 sequences, and train the model for exactly one epoch.

Here we give the details regarding the best-first search strategy used in evaluation: we maintain a priority queue of search nodes with queue length 32. The accumulated log probability of the proof steps is used as the queue priority. For each theorem to prove, we first initialize the queue with one node that has the theorem declared and no proof step applied to it. At each time-step, we deque and sample 32 possible proof steps to apply to the node. The nodes corresponding to steps that successfully proceed the proofs then get added to the queue. We repeat this process until the theorem is successfully proven, or we reach our computational budget.

**Machine specification**  For pre-training, fine-tuning, and evaluation, we use a TPUv3 with 8 cores from Google Cloud Platform. The Isabelle process has access to up to 32 CPU cores. We estimate that running all the experiments in this paper requires a total of 780 TPU hours.

Table 3: Proof success rates on miniF2F.

| Model | valid | test |
|---|---|---|
| PACT [22] | 23.9% | 24.6% |
| FMSCL [36] | 33.6% | 29.6% |
| Ours | **37.3**% | **35.2**% |

## 5.3 Result

We use Codex with greedy decoding to formalize 3908 mathematical problems in `algebra`, `inter-mediate algebra`, and `number theory` from the training set of MATH [24], with the same few shot prompts used in Section 4.4. Out of them, 3363 of the autoformalized theorems are syntactically correct. We then perform expert iteration on this dataset.

We start with a neural theorem prover ($M_0$) as described in Section 5.2. In our first iteration, $M_0$ proves 782 theorems, with a success rate of 23.3% (out of 3363). This gives us a new set of verified proofs to further train the neural theorem prover. We proceed to fine-tune our neural theorem prover in the fashion described in Section 5.1 to get a new prover ($M_1$). This process is repeated in the second iteration, giving us 1011 successful proofs from the autoformalized theorems (30.1%). We fine-tuned $M_0$ again on all available deduplicated proofs to obtain $M_2$.

After each stage of fine-tuning, we evaluate the neural theorem prover on miniF2F [52]. The results are shown in Table 3. The base model ($M_0$) has a success rate of 28.3% and 29.9% on the validation and test fractions of miniF2F respectively. It can be observed that the first expert iteration increases the success rate of the neural prover by 7.8% and 4.1% to 36.1% and 34.0% on the valid and test sets. The second iteration further improves them both by 1.2%, to 37.3% and 35.2%. By doing two expert iterations on the autoformalized theorems, the neural prover achieves a success rate that is 5.6% higher than the previous state-of-the-art.

## 6 An Outlook on Autoformalizing Advanced Mathematics

So far, we focused on mathematical competition problems, in which we achieve significant results using autoformalization. Not only can LLMs autoformalize non-trivial theorems, the autoformalized theorems can also improve neural prover performance. In this section, we take a peek into more advanced mathematics. We hope to identify some of the limitations of our methods when it comes to autoformalizing mathematics in the wild.

### 6.1 Autoformalization: from Natural Language to Isabelle

Autoformalization is extremely challenging in the sense that the model needs to (1) bridge the logical gaps left in pen-and-paper proofs, (2) assume the implicit contexts and assumptions, and (3) align informal definitions/concepts to formal ones. This task is further complicated considering the context can change from time to time and the same mathematical object can be formalized in subtly different ways. In this paper, we only deal with autoformalization of theorem statements, where the model is mainly challenged by definition alignments. Compared to competition problems in Section 4.4, advanced mathematical statements assume more context information that additionally obscures the alignment.

Figure 3 shows a typical case where the model fails to align the informal definition to the formal one. Based on the notation used, the model does not correctly distinguish between products of numbers, products of sets, and products of topological spaces. In Isabelle, the correct conclusion for the statement should be `compact_space (product_topology X I)`, where $I$ is an index set that ought to have been introduced in the `fixes` and `assumes` sections.

### 6.2 Informalization: from Isabelle to Natural Language

So far, we explored one direction of translation: from natural language statement to formal statement in Isabelle. The other direction, *informalization*, is also of great importance for two reasons: (1) the informal texts are much easier for humans to comprehend and communicate, and (2) we can

> **Question:** *An arbitrary product of compact spaces is compact.*
>
> ---
>
> **Codex Output:**
> ```
> theorem
>    fixes X :: "'a set set"
>    assumes "\<forall>x \<in>X. compact x"
>    shows "compact (\<Prod> x \<in> X. x)"
> ```

Figure 3: A formalization for an advanced mathematical statement by Codex.

> **lemma** `seteqI:`
> ```
> "⟦⋀x. x∈A ⟹ x∈B;
> ⋀x. x∈B ⟹ x∈A ⟧
> ⟹ A = B"
> ```
>
> Translate the Isabelle version to a natural language version:
>
> ---
>
> **Codex Output:**
> "If $A$ and $B$ are sets such that $A \subseteq B$ and $B \subseteq A$, then $A = B$."

> **lemma** `topologyI:`
> ```
> "⟦ ⋀ x y. ⟦ is_open T x; is_open T y⟧ ⟹
> is_open T (x ∩ y);
>     ⋀ M. ∀ m ∈ M. is_open T m ⟹
> is_open T (⋃ M)
>     ⟧ ⟹ topology T"
> ```
>
> Translate the Isabelle version to a natural language version:
>
> ---
>
> **Codex Output:**
> "If $T$ is a set and $T$ is closed under finite intersections and arbitrary unions, then $T$ is a topology."

Figure 4: Two perfect translations from Isabelle code to natural language by Codex.

align translated informal statements with formal ones to create data, and use the back-translation techniques [41] to potentially boost the translator's performance further. In this section, we explore Codex's capability of translating formal Isabelle statement to natural language.

A corpus of 38 formal-language theorems, lemmas, and definitions is selected by an Isabelle expert. These statements are automatically translated to informal mathematics using Codex; to see the prompt we used and the results for all 38 examples, see Appendix B.3 and E.2. We present two examples of informalization in Figure 4. Of the 38 examples, 36 were translated to a reasonably coherent statement, and 29 of these statements (76%) were more-or-less correct, giving a vastly better success rate than the 25% success rate of formalization (Section 4.4). Our main conclusion is that for advanced mathematics, the model is better at informalization than formalization, showing the prospect of backtranslation style algorithms.

Note that the standard is more relaxed here since we assume a human reader will supply the obvious context and correct mistakes when the intended meaning is obvious (intended by the hypothetical human writer of these sentences). To illustrate, an example of a minor "acceptable" error: assuming that "$w, z$ are in the same connected component of the plane" when, in context, it is clear that $w, z$ should be assumed to be in the same connected component of the complement of a previously specified curve. (The assumption as originally stated is trivial.) For an example of a major error: almost-perfect translation of the Central Limit Theorem that omits the assumption of identical distributions.

## 7   Discussion

**Promise of Autoformalization with LLMs**   We have seen that automated formalization of informally given natural language statements is generally possible, even with language models not trained for this particular task. Also, automatically formalized statements are useful for training and improving the reasoning capabilities of automated neural provers. Our hope is that improved versions of this methodology will be capable of enabling a positive feedback loop involving formalization and formal reasoning that has the potential of reaching human level capabilities in both respects, as was suggested by [42].

**Limitations and future directions** We use a static model for the formalization process. For large-scale autoformalization, we will need to formalize larger theories, preferably without fine tuning the model, as training it could be cumbersome and resource consuming. However, in order to utilize the newly added notions, the model would need to keep whole large theories in the current context window, which exceeds those of the current LLMs. This limits our approach to the generation of fairly small pieces of formal mathematics and the automatic formalization of entire theories including their definitions will require new research ideas. One path towards this goal might be the use of continuous training or expert iteration, cycle-consistency-based training [30, 46], or novel uses of in-context learning. To generate larger theories we will also need neural networks that can recall longer sequences (current LLMs are typically limited to a few thousand words). Retrieval-augmented language models, such as the memorizing transformer [51] offer one path to overcome this limitation.

**Societal Impact** While the potential of creating negative societal impact through formalizations is small, the use of LLMs always comes with risks. For example, for deploying an autoformalization tool using LLMs we would need to consider the inclusivity of variable and lemma names, and of the attribution of scientific ideas.

## Acknowledgement

AQJ is supported by a Peterhouse Graduate Research Studentship. WL is supported by the ERC Advanced Grant ALEXANDRIA (Project GA 742178).

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
