# OpenReview forum: "Autoformalization with Large Language Models"
_NeurIPS.cc/2022/Conference — NeurIPS 2022 Accept_

### Official Review · Reviewer_XKbt · 2022-07-11

**Rating:** 6
**Confidence:** 4
**Soundness:** 3 good
**Presentation:** 4 excellent
**Contribution:** 3 good

**Summary:**

In this work, the authors use large language models, specifically PaLM and Codex, to automatically formalize natural language text into Isabelle. In addition, the authors demonstrate that training a previous neural theorem prover on these automatically formalized theorems increases the success rate, achieving a new SoTA on miniF2F.


**Questions:**

- You give a detailed analysis of the failure modes of your system. Given this, is it possible at all to predict (or assign a probability) when your formalization of a natural language prompt will be correct?
- In order to learn problems of a given domain, my understanding is that it needs to be trained on problems in that domain. In this sense, it's not clear to me how much overfitting is being done here. Could the authors elaborate on this?


**Limitations:**

Yes

**Strengths And Weaknesses:**

Strengths:
- The paper provides a new and interesting application of LLM’s: automatically formalizing text into theorems. I find that this is an important direction of research: research in automatic formalization is scarce, and there is not much data for formalizations in Isabelle, Lean, Coq, etc.
- Even just using Codex out of the box, the results are very impressive, showing that LLM’s are able to generalize beyond their training data to unseen tasks. This work will likely be a starting point and inspire future work and ideas in the area.
- The authors do a detailed human evaluation of failure cases, helping us understand when Codex can fail to generate correct programs.
- The authors provide an analysis, in the appendix, on the effect of LLM scaling, which shows the benefits of scaling.

Weaknesses:
- Of the 150 problems, Codex is only able to translate 38 problems perfectly. Thinking of the use cases of LLM’s, it can be tedious and time consuming for users to verify if a generated output is correct (see https://tianyi-zhang.github.io/files/chi2022-lbw-copilot.pdf). As with systems such as Copilot, generating an almost-correct formalization of an input statement, for example one that compiles but is missing a minor detail and can be quite dangerous. Therefore, I believe it is important to include a way to verify (with reasonable accuracy) if a generated formalization indeed matches the specification. The authors also acknowledge, at the end of Section 6, that almost-perfect translations may occur. Relatedly, the use case of the system described in Section 5.1 relies on being able to identify the set of successful proofs S_i, which the paper does not include.
- I feel that a one-shot approach is not particularly fitting for a problem of this complexity. Rather, I feel that a more compositional approach of breaking down the formalization problem into smaller pieces is more appropriate here. For example, in Case Study 2, the model might first pick up on the “line up in rows of eight, there are seven left over” piece and formalize that, and then continue with the rest of the prompt.

---

> ### Author Response · Authors · 2022-08-02
> **Thank you for your encouraging reviews. We address your questions below.**
>
> We thank the reviewer for your encouraging comments!
>
> **“It is important to include a way to verify (with reasonable accuracy) if a generated formalization indeed matches the specification.”**
>
> We very much agree with the reviewer that a way to verify will be great to have. However this is a difficult problem as it requires both natural language understanding (to understand the natural language specification) and formal language understanding. It is a big topic of research in its own right.
>
> **“Relatedly, the use case of the system described in Section 5.1 relies on being able to identify the set of successful proofs S_i, which the paper does not include.”**
>
> Proof assistant Isabelle automatically verifies the successfulness of the proof. We will add more details to clarify this in the paper.
>
> **“I feel that a one-shot approach is not particularly fitting for a problem of this complexity.”**
>
> This was our intuition too, which is why we were so surprised that this approach worked. Nevertheless, we believe what the reviewer suggested could be a way to improve upon the existing work, and we leave it to future work.
>
>
>
> **“Is it possible at all to predict (or assign a probability) when your formalization of a natural language prompt will be correct?”**
>
> We have not studied this in our experiments for this paper, but think this is a fascinating idea. Thanks to the reviewer for such an interesting idea. We will leave this to future work.
>
>
> **“In order to learn problems of a given domain, my understanding is that it needs to be trained on problems in that domain. In this sense, it's not clear to me how much overfitting is being done here.”**
>
> We are not clear which problems you refer to by “learn problems of a given domain”. We can see two interpretations: 1. Learning how to autoformalize, 2. Learning how to prove a new theorem with the neural theorem prover. For the former, we do not “train” the model by updating the weights of the model, but by simply providing a few examples in the prompt. Learning from only a few examples and quickly generalizing to those is not considered as overfitting. As for the latter, our results show that by training on autoformalized theorems, the model improves its capability in proving the theorems in the same domain. The neural prover may overfit to this distribution after training on theorems of the same domain, but we can easily regularize it by co-training with all the proofs from all domains.

---

### Official Review · Reviewer_4XAX · 2022-07-11

**Rating:** 4
**Confidence:** 4
**Soundness:** 1 poor
**Presentation:** 2 fair
**Contribution:** 2 fair

**Summary:**

The paper evaluates the ability of pre-trained large language models such as Codex and PaLM on the problem of auto-formalizing natural language mathematics to formal specifications. The paper also shows how the model-generated formal specifications can be used to improve a neural theorem prover iteratively using expert iteration algorithms.

**Questions:**

The details about the neural theorem provers in section 5.2/5.3 are confusing.
- What is Sledgehammer and what is the purpose of invoking it?
- It is not clear what is the $M_0$ model here --- is it the LISA model or GPT-2 style decoder that is pretrained on the Pile.
- Why the last fine-tuning step in lines 268-269 in Section 5.3?

**Limitations:**

Limitations are addressed.


**Strengths And Weaknesses:**

STRENGTHS:

The paper does a good job of motivating the autoformalization problem. The case studies in this paper allow us to get some preliminary insight into the challenges of this problem and the capabilities of existing LLMs.

The idea of using autoformalized theorems to generate a dataset of natural theorems and using them with the expert iteration algorithm to improve a neural theorem prover is well executed and the results are impressive with almost a 5% performance improvement.

WEAKNESSES:

The main disappointment with the paper is the lack of a rigorous evaluation. Most of the insights presented in this paper are gathered from just a few small case studies or a small-scale user study. The paper does not thoroughly explore various baselines/ablations. More details specifically,
   * For table 1, why not also include PaLM Coder model to make it a fair comparison to Codex which is trained on more formal code? Also, it will be useful to look at the performance of some publicly available models such as GPT-J.  Table 1 should also report other metrics such as CodeBleu since Bleu is not always the best metric for code.
   * The paper infers the need for few-shot prompting from just a single task case study, but there are no ablations with respect to the number of few shot examples for the results in table 1.
   * The neural theorem proving experiments also lack several baselines such as (i) using various different models for autoformalization (ii) temperature sampling/non-greedy decoding for autoformalization to collect more theorems (since we do not really need perfect autoformalization for these experiments) (iii) randomly sampled formal theorems from a DSL.
   * Section 6 is very hand-wavy and again derives insights from a few case studies. Why not do Isabelle to Natural language on the miniF2F dataset and report Bleu scores for various models/ablations?

Overall, the contribution of this paper is very thin --- the paper neither produces new techniques nor produces new datasets for training/evaluation nor does a thorough evaluation. So, in my opinion, this paper does not pass the bar for a NeurIPS paper.

---

> ### Author Response · Authors · 2022-08-02
> **Thank you for your reviews. We address your concerns below. Part I.**
>
> We thank the reviewer for your detailed reviews. We address your concerns below.
>
> **“Novelty”**
>
> We want to stress that the fact that autoformalization can be addressed at all without any aligned data of informal and formal statements is very surprising and novel, which is the main observation presented in this paper. In terms of technical innovation, we devised a new evaluation methodology that measures the impact of autoformalization by its ability to improve neural theorem provers, also we found that surprisingly few theorems were necessary for a successful few-shot promoting of the task  We will clearly mark these contributions in the introduction.
>
> **“Other LLMs for comparisons, such as PaLM Coder, GPT-J.”**
>
> The question which LLMs perform best (and how to improve their autoformalization performance) is extremely interesting, but out of scope for this work. The main point of this paper is that autoformalization is possible with language models in the first place - even without aligned data.
>
> **“Table 1 should also report other metrics such as CodeBleu since Bleu is not always the best metric for code.”**
>
> The suitability of CodeBleu for proofs has not been demonstrated. CodeBleu requires parsing of the statements and the extraction of abstract syntax tree and is, hence, considerably more complicated. Further, parsing Isabelle proofs is highly complicated, as it supports user-defined syntax/commands, i.e. the AST can change after some theory files are loaded. For Table 1 we opted for the simplest metric that gives at least some indication of the relative performance of the models. Please note that we evaluate the usefulness of the autoformalized statements separately from Table 1 by studying their impact on neural theorem provers (Section 5).
>
> **"Section 6 is very hand-wavy and again derives insights from a few case studies. Why not do Isabelle to Natural language on the miniF2F dataset and report Bleu scores for various models/ablations?"**
>
> Section 6 aims to provide a first exploration into autoformalization in advanced mathematics. For this section, we also do not have a proper benchmark like for mathematical competition problems (miniF2F). Hence we could mostly derive insights from our case studies. Notably, we performed a human evaluation on 38 examples for informalization for advanced mathematical statements. They provide solid evidence that this direction is promising.
>
> On the other hand, we did not do informalization for miniF2F, because there are many ways to informalize a formal mathematical competition statements. Consider the example in Case Study 2, an equation could have many ways of interpretation (informalization). Hence computing the BLEU scores makes little sense here.
>
> **"The paper infers the need for few-shot prompting from just a single task case study, but there are no ablations with respect to the number of few shot examples for the results in table 1."**
>
> We thank reviewers for the suggestion of this ablation. We will include this ablation in our final version.
>
> **"The neural theorem proving experiments also lack several baselines such as (i) using various different models for autoformalization (ii) temperature sampling/non-greedy decoding for autoformalization to collect more theorems (since we do not really need perfect autoformalization for these experiments) (iii) randomly sampled formal theorems from a DSL."**
>
> We thank the reviewer for raising these baselines. i) As we mentioned previously, we do not focus on LLM comparisons in this paper. ii) it is a good idea but we will leave exploration of inference-time techniques (different decoding techniques) to future works. iii) Random sampling of non-trivial true theorems is difficult and a research topic on its own. The only work we are aware of that performs random sampling for an interactive theorem prover is INT [Wu et. al.], and recently adapted to Lean in FMSCL [Polu et. al.]. The main limitation of this kind of work is that the theorems that got sampled are very limited to the DSL one designs, and it requires significant engineering effort to expand the DSL. Future advancements are required to make this approach scalable, and hence deserves a research topic on its own.
>
> **References**
>
> Wu et. al., INT: An Inequality Benchmark for Evaluating Generalization in Theorem Proving.
>
> Polu et. al., Formal Mathematics Statement Curriculum Learning.

---

> > ### Author Response · Authors · 2022-08-02
> > **Thank you for your reviews. We address your concerns below. Part II.**
> >
> > **Questions about neural prover training: “What is Sledgehammer and what is the purpose of invoking it?”
> > Sledgehammer is a push-button automatic proving method that is based on state-of-the-art (symbolic) automatic theorem provers. It can be invoked as a comparison/baseline for a neuro-proving method (e.g., based on large language models).
> > “It is not clear what is the “M0” model here --- is it the LISA model or GPT-2 style decoder that is pretrained on the Pile.” “Why the last fine-tuning step in lines 268-269 in Section 5.3?”**
> >
> > We thank the reviewer for raising these questions. The settings regarding sledgehammer and prover set up are identical to those of a concurrent submission [Anonymous, 2022, added in the supplementary material], which achieves the SOTA results without additional training data on the MiniF2F dataset: The $M_0$ model is the LISA model that was used by Thor. Sledgehammer is a tool that helps selecting useful premises for the language model, which the Thor work found to be helpful for theorem proving. We have updated the paper in Section 5.2 to justify using the same settings as Thor.
> >
> > The fine-tuning step in lines 268-269 denotes the second expert iteration, i.e. the process of obtaining $M_2$ from $M_0$. It refers to the same process introduced in Section 5.1. We realize that our description of it is ambiguous and have updated the paper to point out what the fine-tuning step refers to.
> >
> > **References**
> >
> > Anon Anonymous.  Thor:  Wielding hammers to integrate language models and automated theorem provers. 2022.

---

> > > ### Comment · Reviewer_4XAX · 2022-08-06
> > > **Follow-up**
> > >
> > > Why is $M_2$ finetuned from $M_0$ rather than from $M_1$?
> > >
> > > I still don't understand the purpose of pretraining and finetuning mentioned in lines 247 to 258 in the revised paper.

---

> > > > ### Author Response · Authors · 2022-08-06
> > > > **Addressing your followup questions.**
> > > >
> > > > Thank you for asking those questions! We will improve our paper by incorporating our answers into the paper.
> > > >
> > > > **"Why is $M_2$ finetuned from $M_0$ rather than from $M_1$?"**
> > > >
> > > > This is a common technique to avoid overfitting. Continue finetuning the same model on a small dataset may easily lead to overfitting. See it was also used in FMSCL [Polu et. al.], STaR [Zelikman et. al.].
> > > >
> > > > **"purpose of pretraining and finetuning"**
> > > >
> > > > Language modeling pre-training is found to be useful for neural theorem proving, from the seminal work of GPT-f [Polu & Sutskever]. We hence follow this when training our neural theorem prover. Finetuning is performed on the Isabelle theorem proving dataset.
> > > >
> > > >
> > > > **References**
> > > >
> > > > Polu et. al., Formal Mathematics Statement Curriculum Learning.
> > > >
> > > > Zelikman et. al., STaR: Bootstrapping reasoning with reasoning.
> > > >
> > > > Polu & Sutskever, Generative Language Modeling for Automated Theorem Proving

---

> > ### Comment · Reviewer_4XAX · 2022-08-06
> > **Follow up**
> >
> > Thank you for your response.
> >
> > I want to clarify that the purpose of asking to implement and evaluate the above suggested baselines is not to suggest that those baselines would perform better than your approach, but to properly inform the reader about the impact of the various design choices you had to make. For e.g. since showing that LLMs can achieve auto formalization without any extra training/data is an important contribution of your work, it is the responsibility of the authors to establish what aspects of LLMs are crucial to enable this ability by evaluating on several different LLMs. Moreover, since the paper only used Codex and Palm which are not publicly available, it is hard for an average reader to establish the validity of this paper. Similarly, I feel it is very important to evaluate different various of inference-time decoding techniques to properly establish the other contribution mentioned by the authors. For a paper that does not have a lot of technical contribution, I would at least expect a detailed evaluation for a big conference such as NeurIPS.

---

> > > ### Author Response · Authors · 2022-08-06
> > > **Thank you for the followup.**
> > >
> > > **"properly inform the reader about the impact of the various design choices"**
> > >
> > > We agree with the reviewer that it is definitely important to understand what the impacts are. However, this is a very difficult question to answer given how little of our understanding is for pre-training in general. We believe the comparisons we made in the paper is our best shot at answering this question. By comparing Codex with PaLM, we learned formal data did improve model's autoformalization capabilities. By comparing different model sizes of PaLM, we learned autoformalization scales well with model sizes. We do not feel the need to compare further with those model suggested by the reviewer, because it is hard to tell what extra insights we can obtain from those comparisons.
> > >
> > > **"since the paper only used Codex and Palm which are not publicly available, it is hard for an average reader to establish the validity of this paper."**
> > >
> > > We thank the reviewer for the suggestion. We agree with reviewer on this, and will include a public model, such as GPT-J in the final version of the paper.
> > >
> > > **"evaluate different various of inference-time decoding techniques to properly establish the other contribution mentioned by the authors"**
> > >
> > > We thank the reviewer for the suggestion. We do not think studying inference-time techniques is required to establish our contribution. We will leave this to future work.
> > >
> > > **"For a paper that does not have a lot of technical contribution, I would at least expect a detailed evaluation for a big conference such as NeurIPS."**
> > >
> > > We would like to re-emphasize the contribution of the paper.
> > >
> > > The technical contribution of the paper is to establish the insight that the ability of autoformalization can emerge naturally from language modeling, without any aligned data of informal and formal statements. This opens an entirely new direction of research, and has unlocked many abilities that we previously can only dream of. This has already inspired many followup works, and also was picked up by the formal community (https://marketplace.visualstudio.com/items?itemName=hoskinson-ml.lean-chat-vscode), where Lean users now can chat with Codex to generate definitions, lemmas. The official mathlib library (Lean's formal library) documentation will be annotated with natural language description, generated by informalization from Codex.
> > >
> > > In addition, we devised a new evaluation methodology that measures the impact of autoformalization by its ability to improve neural theorem provers. We also found that surprisingly few theorems were necessary for a successful few-shot promoting of the task, and strong correlation of the examples in the prompt and the autoformalization ability.

---

### Official Review · Reviewer_BiQy · 2022-07-12

**Rating:** 8
**Confidence:** 5
**Soundness:** 4 excellent
**Presentation:** 3 good
**Contribution:** 4 excellent

**Summary:**

This paper studies the problem of autoformalization that translates the statements of the math problems in natural language to formal language, and demonstrates how the autoformalized formal problem statements in Isabelle/HOL can be used to improve the performance of theorem provers. The formal statements are generated by LLMs (PaLM and Codex) in the manner of few-shot prompting. They are used in expert iteration to train the neural theorem prover. By manually inspecting the autoformalized statements of 150 problems in the MATH dataset, 38 of them are translated correctly by Codex. Experiments demonstrate that 3363 autoformalized statements of the problems in the Math dataset can effectively improve the pass rate of the prover on the MiniF2F benchmark after expert iteration (from 29.9% to 35.2% after 2 iterations).

**Questions:**

1 Why do you choose Isabelle/HOL for autoformalization compared with HOL Light, Coq and Lean?
2 Could you provide more details about the settings of BFS in expert iteration and evaluation and the training hyperparameters of expert iteration?


**Limitations:**

The limitations of the proposed methods are mentioned in sections 6 and 7. I have no more concerns to discuss.

**Strengths And Weaknesses:**

Strengths:
1 This paper studies the problem of autoformalization, an very important and promising topic in the domain of automated theorem proving.
2 It is a reasonable initial attempt to formalize the informal statements by few-shot prompting LLMs. The performance of this simple approach seems to be quite positive. By manually inspecting a few translated formal statements in the appendix, although most of the translations are not perfect, they capture the overall meanings of the informal statements with misaligned details. I think this also explains why these formalized statements are useful for training the provers.
3 Experiments on MiniF2F validate that the formalized problem statements could be useful training targets for advancing the neural theorem provers.

Weakness:
The settings of BFS in the evaluation on MiniF2F are not mentioned. It is unclear if the comparison with the prior works is proper since different interactive theorem provers are used.

---

> ### Author Response · Authors · 2022-08-02
> **Thank you for your encouraging reviews.**
>
> We thank the reviewer for encouraging comments.
>
> **“Why do you choose Isabelle/HOL for autoformalization compared with HOL Light, Coq and Lean?”**
>
> The miniF2F benchmark only contains a complete set of Isabelle and Lean statements. Hence we did not start with other choices. Out of Isabelle and Lean, we chose Isabelle for two reasons: 1. We have Isabelle experts in our team who can help to perform human evaluations, 2. We have a working infrastructure to run neural theorem provers for Isabelle but not for Lean.
>
> **“The settings of BFS in the evaluation on MiniF2F are not mentioned. It is unclear if the comparison with the prior works is proper since different interactive theorem provers are used.” “Could you provide more details about the settings of BFS in expert iteration and evaluation and the training hyperparameters of expert iteration?”**
>
> We thank the reviewer for pointing out the incompleteness of the settings and hyperparameters. We have duly updated the paper and added these contents at the end of Section 5.2.
> One concurrent submission, Thor [Anonymous, 2022] (attached as a supplementary material), achieves SOTA results (without additional training data) on the MiniF2F dataset, with the Isabelle interactive theorem prover. We use Thor as the base theorem prover model ($M_0$) and make significant performance improvements over it with expert iteration. Therefore we believe the claim that expert iteration with autoformalized examples improves model performance is fair, given that the two works use the same interactive theorem prover.
>
> Anon Anonymous.  Thor:  Wielding hammers to integrate language models and automated theorem provers. 2022.

---

### Official Review · Reviewer_UgXb · 2022-07-25

**Rating:** 6
**Confidence:** 3
**Soundness:** 3 good
**Presentation:** 4 excellent
**Contribution:** 2 fair

**Summary:**

This paper studies the ability of LLMs to automatically translate mathematical statements (written in natural language) to formal statements in the language of a theorem prover (in this case, Isabelle/HOL). The authors also demonstrate the usefulness of such a translation system by incorporating it in an expert iteration procedure for training a neural theorem prover. The main contributions are as follows.

+ Few shot learning experiments demonstrating that LLMs (Codex, in particular) can autoformalize a large fraction (25.3%) of mathematics competition problems in Isabelle.
+ Case study characterizing incorrect formalizations among 150 problems formalized by Codex.
+ A neural theorem prover trained using expert iteration with the help of theorems autoformalized by Codex that achieves state-of-the-art proof rate.

**Questions:**

* It is mentioned that FMSCL baseline [35] uses human written formal statements for expert iteration. Is it possible to generate formal statements at random or train a statement generator for this purpose? Since many of the autoformalized theorems may not actually correspond to the problem statement in the dataset, it might be worth comparing to randomly generated theorems for expert iteration.
* It is mentioned that the amount of Isabelle data used for training is a very small fraction and it is likely that some of the syntax is learned from the few-shot examples. In order to understand if that is the case, did you consider modifying the Isabelle syntax slightly, such as changing "shows" to "prove" etc., and repeating the few-shot experiments (at least some of them)?

**Limitations:**

Limitations w.r.t. advanced math statements and scalability are discussed in the paper.

**Strengths And Weaknesses:**

### Strengths

+ __The paper is well written and is easy to follow.__ Lots of examples are provided to illustrate the autoformalization capabilities of LLMs as well as the failure cases. In particular, I found the insights about teaching the concept of a linear function quite fascinating.
+ __The problem studied is interesting and relevant.__ The ability to autoformalize statements is very helpful in reducing human effort required to formalize paper proofs in a proof assistant such as Isabelle.

### Weaknesses

- __Limited novelty in terms of methodology.__ Although the application here is different, the methods used for few shot learning using LLMs as well expert iteration for training neural theorem provers have been studied before. However, it is interesting to see that these ideas can be successfully applied in this context.
- __Experiments limited to mathematics contest problems.__ It was good to see this mentioned in the paper. It would be interesting to see how few shot learning helps with autoformalization of theorems about safety-critical software or complex theorems requiring understanding of many math concepts as, according to me, these appear to be the cases where autoformalization would be most valuable. Achieving autoformalization of a statement such as the one in Figure 3 would also require new methods for incorporating understanding of concepts such as products of spaces.
- __A few unexplained choices in the experiments.__ As far as I understand, a single prompt (with two fixed examples) was used for all the few shot experiments. Although it was random, it would have been interesting to see the results for a few choices of training examples. The proof rate of the neural theorem prover is measured using greedy decoding, however, the baseline FMSCL seems to achieve better pass@8 performance. It would be nice to see if pass@8 performance improves as well.

---

> ### Author Response · Authors · 2022-08-02
> **We thank the reviewer for your encouraging comments and thought-provoking questions. We address your questions below.**
>
> We thank the reviewer for your encouraging comments and thought-provoking questions. We address your questions below.
>
> **“Limited novelty in terms of methodology”**
>
> We want to stress that the fact that autoformalization can be addressed at all without any aligned data of informal and formal statements is very surprising and novel, which is the main observation presented in this paper. In terms of technical innovation, we devised a new evaluation methodology that measures the impact of autoformalization by its ability to improve neural theorem provers, also we found that surprisingly few theorems were necessary for a successful few-shot promoting of the task  We will clearly mark these contributions in the introduction.
>
> **“Experiments limited to mathematics contest problems”**
>
> The paper focuses on mathematical competition problems as the first proof-of-concept of this idea. In Section 6, we started looking into more advanced mathematics, and showed several examples that illustrate the potential challenges ahead. However, this doesn’t mean these models fail at them entirely. In fact, given how promising the results are in “informalization”, we believe some training with back-translation can significantly improve the quality of the formalization for these advanced mathematical statements. We are actively working on extending our work to problems beyond mathematical competition problems, and we believe this work is an important milestone towards this goal.
>
> **“Choices of prompts”**
>
> In Section 4.3, we randomly sampled 10 examples to construct the prompt. As per reviewer’s suggestion, we did this 3 more times, and found the variance of BLEU scores were quite low. The BLEU scores we got were: 55.23, 54.32, 57.97. We will include this new result in the paper.
>
> On the other hand, as we showed in “Case Study 3”, we believe the examples in the prompt can have a big influence on the results. Once we include an example that explains the concept of a line, the model can then formalize the Case Study 3 perfectly (see appendix B.3 for the example). We however did not perform prompt engineering in great detail, as it is not the main focus of the work.
>
> **“Generating formal statements at random or training a theorem generator?”**
>
> Random sampling of non-trivial true theorems is difficult and a research topic on its own. The only work we are aware of that performs random sampling for an interactive theorem prover is INT [Wu et. al.], and recently adapted to Lean in FMSCL [Polu et. al.]. The main limitation of this kind of work is that the theorems that got sampled are restricted to the DSL one designs, and it requires significant engineering effort to expand the DSL. Future advancements are required to make this approach scalable, and hence deserves a research topic on its own.
>
> As for training a theorem generator, the main problem is that the model is not capable of generating theorems outside of its training distribution (existing formal library) (see, e.g., skip-tree [Rabe et. al.]). In this case, we do not expect the model to generate the kind of mathematical competition statements we autoformalized, as they do not appear in the existing Isabelle formal library. This is exactly the main point of doing autoformalization – that it allows one to translate entirely novel distributions of theorems from natural language mathematics.
>
> **“Pass@8”**
>
> Thanks reviewers for suggesting this metric. We have started evaluating. Since it is expensive to run, we do not expect to finish the experiments before the rebuttal period ends. We however will include this result in our final version of the table.
>
> **“Did you consider modifying the Isabelle syntax slightly, such as changing "shows" to "prove" etc., and repeating the few-shot experiments (at least some of them)?”**
>
> That is a very good idea! We tried the suggestion given by the reviewer: we modified Isabelle syntax slightly by changing “shows” to “prove” in the prompt, and used Codex and PaLM (540B) to formalize the problem in the case study 1. Surprisingly, the two models behave differently. Codex did not pick up on the fake syntax, and still outputted the correct Isabelle code, whereas PaLM picked it up and changed “shows” to “proves” in its proof.
>
> This may be due to the fact that Codex was potentially trained on the formal data more frequently compared to PaLM. In general, we also observed that Codex learns Isabelle syntax better than PaLM, which is aligned with this observation.
>
> We found this an interesting observation and will include more details in the appendix in our future version.
>
> **References**
>
> Wu et. al., INT: An Inequality Benchmark for Evaluating Generalization in Theorem Proving.
>
> Polu et. al., Formal Mathematics Statement Curriculum Learning.
>
> Rabe et. al., ​​Mathematical Reasoning via Self-supervised Skip-tree Training.

---

> > ### Comment · Reviewer_UgXb · 2022-08-08
> > **Rebuttal Reply**
> >
> > Thanks for the detailed comments. I appreciate the new experiments and think they improve the validity of the claims. However, it appears a large part of Isabelle syntax is learned from pretraining data and hence would encourage the authors to rewrite lines 141-145 while taking into account the results of the new experiments.
> >
> > Furthermore, although the neural theorem proving experiments are interesting, I don't believe one can claim that they measure the impact of autoformalization since autoformalized theorems may be incorrect. These experiments show that using LLMs in a specific way for data augmentation helps but do not measure the autoformalization capabilities of these models. I think that this should be clarified in the paper.

---

> > > ### Author Response · Authors · 2022-08-09
> > > **Thank you for your reply.**
> > >
> > > **"encourage the authors to rewrite lines 141-145 while taking into account the results of the new experiments"**
> > >
> > > In line 141-145, we did not claim that the model learned the syntax from the few-shot prompts. Our claims emphasize the fact that there is a small amount of Isabelle proofs on the internet and it is fascinating that Codex can write syntactically correct code. This is surprising if Codex was only finetuned on Github as how it was described in their paper [Chen et. al.], without specializing to any particular language (e.g. Isabelle). Our new experimental results actually strengthens this claim as they show how well Codex remembers Isabelle syntax.
> > >
> > > **"I don't believe one can claim that they measure the impact of autoformalization"**
> > >
> > > We agree with the reviewer that the amount of improvements for neural theorem proving cannot be directly used to compare the autoformalization abilities of different models, as the improvements depend on many factors, such as the initial proving abilities of the neural prover. We will clarify this in the paper. Thanks for the suggestion!
> > >
> > > **References**
> > >
> > > Chen et. al., Evaluating Large Language Models Trained on Code.

---

### Author Response · Authors · 2022-08-02
**Paper & supplementary updated.**

We thank the reviewers for their thoughtful reviews and great feedback. We are especially happy that the reviewers agree that the problem we studied is important and interesting.

We have since updated the paper according to reviewers' suggestions, also included a reference paper in the supplementary material. We look forward to discussing more with you during the discussion period, and hopefully address you concerns.

---

### Meta-Review · Area_Chair_9zsG · 2022-08-24

**Recommendation:** Accept
**Confidence:** Certain

**Metareview:**

The reviewers appreciated the importance of the problem studied. The reviewers had concerns regarding novelty of the proposed solution but were convinced by a detailed empirical evaluation. Overall, this is a clear accept. Congratulations to the authors; please take all the feedback into account while preparing the final version.

**Award:**

No

---

### Decision · Program_Chairs · 2022-09-14

Accept